# Development of a Platform to Align Education and Practice: Bridging Academia and the Profession in Portugal

**DOI:** 10.3390/pharmacy8010011

**Published:** 2020-01-16

**Authors:** Filipa Alves da Costa, Ana Paula Martins, Francisco Veiga, Isabel Ramalhinho, José Manuel Sousa Lobo, Luís Rodrigues, Luiza Granadeiro, Matilde Castro, Pedro Barata, Perpétua Gomes, Vítor Seabra, Maria Margarida Caramona

**Affiliations:** 1Ordem dos Farmacêuticos, Rua da Sociedade Farmacêutica, 18, 1169-075 Lisboa, Portugal; ana.martins@ordemfarmaceuticos.pt (A.P.M.); caramona@ci.uc.pt (M.M.C.); 2Faculdade de Farmácia da Universidade de Coimbra, Polo das Ciências da Saúde, Azinhaga de Santa Comba, 3000-548 Coimbra, Portugal; fveiga@ci.uc.pt; 3Faculdade de Ciências e Tecnologia da Universidade do Algarve, Campus de Gambelas, 8005-139 Faro, Portugal; iramalhinho@ualg.pt; 4Faculdade de Farmácia da Universidade do Porto, Edifício 3, Piso 3. Rua de Jorge Viterbo Ferreira, 228, 4050-313 Porto, Portugal; slobo@ff.up.pt; 5Escola de Ciências e Tecnologia da Universidade Lusófona de Humanidades e Tecnologias, Campo Grande 376, 1749-024 Lisboa, Portugal; Monteiro.rodrigues@ulusofona.pt; 6Faculdade de Ciências da Saúde da Universidade da Beira Interior, Avenida Infante D. Henrique, 6200-506 Covilhã, Portugal; luiza@fcsaude.ubi.pt; 7Faculdade de Farmácia da Universidade de Lisboa, Avenida Professor Gama Pinto, 1649-003 Lisboa, Portugal; diretor@ff.ulisboa.pt; 8Faculdade de Ciências da Saúde da Universidade Fernando Pessoa, Rua Carlos da Maia, 296 4200-150 Porto, Portugal; pbarata@ufp.edu.pt; 9Instituto Universitário Egas Moniz, Campus Universitário, Quinta da Granja Monte de Caparica, 2829-511 Caparica, Portugal; pcrgomes@egasmoniz.edu.pt; 10Instituto Universitário de Ciências da Saúde, Rua Central de Gandra, 1317, 4585-116 Gandra, Portugal; Vitor.seabra@iucs.cespu.pt

**Keywords:** professional practice, education, pharmacy, Competency-Based Education, health-workforce, policy making

## Abstract

Limited fitness for practice may result from a mismatch between education and practice. Aiming to meet the common interests of academics and practitioners, the Portuguese Pharmaceutical Society (PPS) developed the Education and Practice Platform (EPP). The EPP includes one representative from each pharmacy faculty, and all Councils of Speciality Boards of Practice. Brainstorming with involved parties enabled sharing of interests, concerns and identifying a common path. Aims, mission, vision and values were set. The EPP’s mission is to: act as an enabler to foster the quality and adequacy of education through sharing best practices, ultimately leading to facilitate professional integration, and to foster quality development in teaching practices with recognition for autonomy in freedom to teach and to learn. Its vision is an alignment of education and practice with the PPS’ statutes to ensure validation of the competences defined for each practice area, and compliance with international guidance. Key performance indicators (KPIs) were set. Activities developed include the creation of a national forum to discuss education and practice, development of workshops on teaching methods and pharmacy internships, enhanced representation in international events and response to global and national requests. Ongoing work focuses on the creation of a common training framework in hospital and community pharmacy practice adapted to Portugal. The EPP is a worldwide case study, encouraging the development of discussion contributing to an open climate of sharing best practices, indirectly leading to foster a better alignment between education and practice. Many of these results are so far intangible in scientific terms but worth describing.

## 1. Introduction

Pharmacists are health care professionals that may develop their activities in various settings, namely in community pharmacy, hospital pharmacy, the pharmaceutical industry, clinical and genetics laboratories, wholesaling, education and research, to name the most common ones [1]. Within most of these areas, the scope of practice has also been expanding in most countries [2,3]. To be able to practice in these diverse settings, a wide scope of knowledge must be conferred at the entry level during the MSc in pharmaceutical sciences, and then further strengthened during lifelong learning through continuous professional development. However, it has been shown that knowledge per se is insufficient to ensure professionalism, and competence seems to be a better construct to predict fitness for practice [4]. As a result, competency frameworks started emerging nationally, initially in the United Kingdom, and subsequently transformed into global [5]. The first frameworks developed focused on the general levels of knowledge and practice, whereas those subsequently developed aimed for alignment with advanced levels of practice, enabling its use throughout the career [6,7]. Despite the obvious interest of the concept and its useful practical implications, only a few countries have so far been able to incorporate competency development from the initial stages of studentship to advanced phases of the professional career, leading to a fragmented concept [8]. In fact, in many countries, the MSc course is still very much knowledge-oriented, with few practical components both in teaching and in assessment methods, leading to recent graduates’ inability to competently provide pharmaceutical activities in the initial stages of their career [9]. Following the international trends, the PPS initiated work in the development of competency frameworks in 2014. These were focused upon the professional areas of activity, and aimed to identify and define the competences required and those desired within each area of practice [10]. Four frameworks were then developed, including hospital pharmacy, community pharmacy, the pharmaceutical industry and regulatory affairs [11,12,13,14]. A revised version of the regulation for the attribution of pharmaceutical competences has been released in 2018 [15]. However, the extent to which these models are aligned with education at the undergraduate level is perceived to be low [16]. This is not unique in Portugal, but solutions are certainly country and culturally associated. In some countries, policy and legal measures may be the sole solution, whereas in other settings, incentives targeted for academia or for practitioners may be most effective, whereas in other cultures, perhaps debate and aiming for consensual solutions may achieve better results. In other words, there is no single, universal solution to make academia perfectly aligned with practice. 

Aiming to tackle this potential problem, the PPS developed the EPP to meet the common interests of academia and practitioners. This implies that the EPP acts in two parallel fields of action: on the academic side, aiming to influence the innovation in teaching methods; andon the professional side, aiming to collect information about key competences for each practice area that need to be ensured through life-long learning, in order to have a competent pharmaceutical workforce.

## 2. Materials and Methods 

Setting and sample: This project is being developed in Portugal, as part of the plan of activities of the Board of the Portuguese Pharmaceutical Society (PPS), who announced the creation of an Education and Practice Platform (EPP) as one of the priorities for the mandate. Portugal is a country in the south of Europe with roughly 10 million inhabitants, and 15,000 registered pharmacists in 2018. These pharmacists may obtain their degree in one of nine faculties, and then pursue their activity in various areas of the profession, represented by six boards of practice. 

Design: The development of the EPP was initiated by identifying the characteristics of representatives, the president and the short and long-term aims. The EPP’s mission, vision and values were then set by the steering group created during a face-to-face meeting. During this meeting, a brainstorming activity with all involved parties was also organized to share concerns and interests, and to identify a common path. Based on the results of this initial activity, a one-year plan was drawn. 

### 2.1. Narrative Review

A literature search was initially conducted in PubMed (from the date of database launch, being January 1996, to the date the search was conducted, which was June 2019) to explore existing initiatives developed elsewhere that could inspire and support an approach to tackle the identified problem of limited alignment between education (mostly undergraduate) and professional practice. It was decided to use one single database because the intent was merely narrative, and a systematic review was not aimed for. Nonetheless, the Preferred Reporting Items for Systematic Reviews and Meta-Analysis (PRISMA) guidelines (Figure 1) were used. A priori restrictions were not made on the time period or article accessibility, until it could be seen if the number of hits obtained justified imposing any limits. Results were restricted to studies involving human species, languages that could be understood by the researchers (English, French, Spanish or Portuguese) and the type of publication (observational studies, experimental studies and processed information, i.e., reviews and meta-analysis).

The search strategy was developed using the population, intervention, comparator and outcome (PICO) method, and all the keywords selected. Population, intervention and outcome were combined as “Population” AND “Intervention” AND “Outcome”.

The following search strategy was used: ((pharmacy OR pharmacist) AND (education OR academia)) AND (pharmacy practice) AND (alignment OR fitness OR suitability OR adequate OR bridging) AND (competence OR competency) NOT (technician). The main research question to be answered was: Are there teaching experiences at the undergraduate level that better meet the needs of pharmacy practitioners? 

The reviewers (F.A.C. and A.P.M.) assessed independently the publications for eligibility by title and abstract content. In case of disagreement, consensus was reached between three of the authors (F.A.C., A.P.M. and M.M.C.).

### 2.2. Workshop Activities

The 2018 plan included the development of two workshops tackling two of the major concerns of educators and practitioners: teaching methods and internships’ organization. Both activities were organized as a two-fold activity, where the first used a more expositive approach providing the basic concepts and possible alternatives, and the second was organized in a more practical manner aiming to foster self-reflection from all involved.

### 2.3. Involvement in International Events

During the brainstorming, there was a perceived need for the greater involvement of academia in international events focusing on education. The EEP was perceived as a possible means for identification, the prioritizing of events and dissemination of selected ones, whilst encouraging active participation.

### 2.4. Answering to Global Requests 

In line with the limited involvement in international events, most EPP representatives were unaware of global advances in pharmacy education, and as such, the EPP was committed to regularly update all representatives on relevant initiatives, namely to be used as a liaison between national academia and international organizations with a special focus on education, such as the The International Pharmaceutical Federation Education (FIP*Ed*), the European Society of Clinical Pharmacy Education Committee (ESCP EdComm), the European Association of Faculties of Pharmacy (EAFP), to name a few, while also being involved with pharmacy students’ international organizations, such as the European Pharmaceutical Students’ Association (EPSA) or the International Pharmacy Students’ Federation (IPSF). 

### 2.5. Answering to National Needs

Although there is a national framework for education in pharmaceutical sciences, publicly announcing the main features of the degree, the differences in specific topics offered, the organization of teaching activities, internships, MSc theses and post-graduate education, were unmapped. As such, one major feature of the EPP in all activities developed has been sharing and benchmarking, adopting a constructive criticism approach. This means in fact that all participants were encouraged to openly share their curricular details, including not only the main features published in official documents, but also the internal working procedures, including class organization, teaching methods, materials, facilities and by working in this climate, all opened up their own “Pandora’s Box”, gaining space for others to share their own flaws, whilst presenting also different approaches, some of which could be used as ideas to be adapted to improve their weakest points. 

The major key performance indicators (KPIs) set for the EPP are presented in Table 1. These were internally defined by the PPS, but are now being subject to ongoing work to refine and reinforce them. This work is being conducted by a subgroup of academics that are part of the EPP, and have offered to do so.

## 3. Results

### 3.1. Literature Review

The literature search led to the identification of 121 records. After applying the general exclusion criteria, 68 papers were retained and reviewed. Among these, five categories of excluded papers were created: (a) focusing on healthcare professionals other than pharmacists (n = 13); (b) focusing only on education (n = 7); (c) focusing only on practice (n = 11); (d) focusing on satisfaction, expectations or perceptions (n = 9); (e) focusing on the development or validation of survey tools, learning resources or exams (n = 7). Excluding these 47, a total of 21 papers were retained (see Appendix A: Table A1). The literature review did not enable concluding if there are teaching experiences at the undergraduate level that better meet the needs of pharmacy practitioners, mostly given the limited number of publications using controlled trials or other powerful evidence-generating designs. Nonetheless, it is worth highlighting a curriculum reform which was based upon the identification of relevant learning outcomes for various fields of pharmacy professionals. This reform led to the institution of innovative teaching practices cocreated with professionals and students [17]. It is likely that the impact on practice of this initiative will be possible to measure in a few years’ time.

### 3.2. Establishing the Platform

The EPP includes one representative from each of the nine pharmacy faculties, and one representative from the six Councils of Speciality Boards of Practice (CSBP).

Its *mission* is to evaluate the quality and adequacy of education from the perspective of professional integration; to foster quality development in teaching practices with recognition for autonomy in the freedom to teach and to learn.

Its *vision* is the alignment of education and practice with the PPS’ statutes to ensure the validation of the competences defined for each practice area, and compliance with the guidance from international organizations.

Its *values* were defined as “strong links between academia and the profession”; “adequacy to societal demands”; “adequacy of teaching to the new areas of the profession”; and “recognition of the existing human capital”.

A logo was created which represents a flower, which will blossom and become professional. This flower is composed of nine petals, five in green and four in white, representing the two groups of public and private institutions in Portugal (Figure 2).

### 3.3. Workshop Activities

The first workshop focused on teaching models and the adaption of the curricula to become practice-centred and student-centred. An example from Monash University (Melbourne, Victoria, Australia) was presented, where vertical teaching was adopted so that all the topics in the curriculum may interact and their utility for practice may become evident [18]. The methodology booklet produced by EPSA was presented, where students shared their views on the current educational model, while clearly criticizing the expositive approach and the lack of alignment with practice. Following these two activities, participants were mixed, creating groups of faculty members from different institutions, and practitioners and were asked to select one topic area and think which feasible changes they could implement to better tailor their teaching to modern methods, and also to students’ expressed desires.

The second workshop focused on the mandatory final unit which comprises the internships in community or hospital pharmacy, as imposed by the European Pharmaceutical Sciences Graduation Directive, where all faculty members shared their internship plan, how it was organized, evaluated and monitored. Subsequently, practitioners shared their experiences and concerns when confronted with students from different institutions in their practice, either in community or in hospital. In this event, apart from the regular six representatives of the CSBP, where two of them gave their testimony about the concerned areas (community and hospital), representatives from the two pharmacy owners associations and from the Portuguese Association of Hospital Pharmacists were also present. Finally, the experience from students was also shared, while suggesting possible new solutions for a unified and improved integration into the profession.

### 3.4. Involvement in International Events

Since the EPP was initiated, the representation in international events has increased (KPI met). To name a few, an oral communication was presented at FIP summarizing the work here described with greater detail, and in addition a poster was presented at the Innovations in Pharmacy Congress held to celebrate the 100 years of the University of Salamanca (Salamanca, Castile y León, Spain). In 2018/2019, the EPP participated in an international study developed by the ESCP, through which information on the weight of clinical pharmacy-related topics in pre-graduate education in all institutions was provided (Table 2).

### 3.5. Answering to Global Requests

The first activity within this domain developed by the EPP was the adaption of the WDGs, issued by FIP, to Portuguese, once permission was officially granted. A summary brochure with the WDGs was distributed in the National Congress of Pharmacists held in October 2017 to all participants (n = 1500). This brochure served various purposes: (1) to introduce the EPP and explain its intention; (2) to present the Workforce Development Goals (WDGs) developed by The International Pharmaceutical Federation (FIP), and (3) to map the national situation in 2017 (Figure 3).

Once the Report “Transforming Pharmacy and Pharmaceutical Sciences Education in the Context of Workforce Development” was issued in Portuguese by the FIP, the EPP also ensured all members were made aware of it [19].

Further, as the EPP started to become more widespread, it became responsible for providing information to external bodies about education in pharmaceutical sciences in Portugal. As such, the EPP has collaborated in two surveys providing information about the evolution on meeting the WDGs (developed by FIP) and on education and practice in clinical pharmacy (developed by the ESCP).

### 3.6. Answering to National Needs

In 2004, CPD became mandatory in Portugal. Currently, there is a recommendation in place guiding pharmacists to complete 15 CPDs in each cycle of evaluation, which occurs every five years. These recommendations aim to lead pharmacists to better identify their training and education needs according to their area of practice. The PPS is the responsible body for attributing European Credit Transfer System (ECTS) units according to the demands of the course. However, various entities may offer CPD to practicing pharmacists. It was considered to be in the best interest of the regulatory body, of the practicing pharmacists and of the potential providers of CPD, to establish a partnership protocol between the PPS and the nine institutions for the provision of post-graduate education in strategic areas. These areas will be annually identified by the CSBP and conveyed to academia so that they may organize consortia between different schools, eventually including also other professionals, to increase the quality and the diversity of CPD, while ensuring geographical distribution.

During the first activities developed, it became clear that there is limited information about the national education in pharmaceutical sciences. Following the activities, a document was shared among participants with the information collated, allowing everyone to check and eventually correct their own details, whilst having a map of the educative offer in other institutions, both at the undergraduate and postgraduate level. As an outcome of the first event, a summary brochure has been shared among all participants, mapping out the common features of the degrees, while highlighting the diversity captured (Table 3).

## 4. Discussion

To bring academics to the PPS was one aim of the EPP, which was believed to have been clearly achieved with the regular meetings organized. Additionally, it was intended to contribute to a greater proximity between pharmacists’ under- and post-graduation education and professional capacities. This institutional program launched by the PPS in collaboration with all the academic partners for pharmacists’ education in Portugal, and with the representatives of the employers of the various areas of practice, is unique, and enabled greater awareness of academics about current practice, while also opening current educational trends to practitioners. Sitting practitioners from different areas of pharmaceutical sciences, including pharmacy practice, together with academics to share their views and seek common solutions to achieve a stronger workforce, is only a starting point. To achieve clear-cut results, much work is still to be developed. For example, as a result of previous workshops focusing on internships, additional future work will be undertaken to develop a common training framework, both for internships in hospital and in community pharmacy, which will be one of the main KPIs. Other initiatives planned for the near future include bringing patient advocates and representatives to discuss formats of greater collaboration with academia and practice. Although much of the work has been centred on pharmacy practice, the main reason is because it has been perceived as the area needing greater reform or alignment.

Nonetheless, it is worth mentioning that one of the outputs of the first meeting held was a request from the Board of Clinical Analysis and Genetics for additional under- and post-graduate educational offers from academia in the field of pharmacogenetics, an area much more within the more traditional pharmaceutical sciences.

This paper introduces the strategic principles, described through a stepwise methodological approach. The background literature review showed there is some research on the content and format of internships and residency programs (five references), on educational programs to develop specific competences (four references), and also on the identification of educational and training gaps (three references), or even failures in the revalidation (one reference), to name a few of the most relevant. However, there was no evidence found of similar processes or programs undertaken for this purpose. The most similar approach was described by Katajavuori et al. [17], although this focused on only one of the aims of the initiative here described, and furthermore was conducted by one single university, as opposed to conduction at institutional/corporate level, believed to reach a wider target group. However, it is possible that similar initiatives do exist and remain unpublished, mostly because of their policy and less scientific character, making their publication difficult.

The program was developed under the umbrella of the PPS. This is found to be a major strength, as it enabled the commitment from all the Boards of Practice, obviously of the utmost importance for sharing framework practices and current and future healthcare system needs that may be useful to incorporate in curriculum design.

The fact that Portugal is a small country might have helped being able to involve all nine faculties. Nonetheless, experience indicates that this opportunity for frequent gathering to discuss common interests contributed to teambuilding, where academia is a unique team rather than nine teams, all of them interested in contributing to the same goal: better pharmacists for the future.

Furthermore, the option for a stepwise approach enabled building a shared vision, and a common path to meet the needs of the future. The regular discussion of practices, taking into account international and national experiences, standards and guidelines is an additional strength of the project itself.

Many of the educational mismatches with practice identified so far are perhaps not new. However, the fact that they were identified by all, using an open forum process, where everyone’s views were considered equal, including teachers, students and practitioners, was crucial for assuming shared commitments for change.

As an example, the need for more clinical pharmacy training is cited, a shared concern by academia, hospital pharmacists and community pharmacists. The need to innovate in clinical pharmacy has long been debated, and the creation of practice-partnerships has been proposed as a possible solution, whilst recognizing the difficulties of putting this idea into practice, sometimes due to cultural or contextual barriers [20]. Data shown has highlighted the diversity in clinical pharmacy, not only in terms of quantitative weight in the curricular structure, but also in terms of the practical teaching methods adopted. Certainly, there are limitations to the analysis shown, namely the fact that optional topics have been ignored, and some faculties have a strong offer in clinical pharmacy provided by such topics (e.g., hospital pharmacy). There is also the possibility of misclassification, as some topics have names that may or not fit into the definition of clinical pharmacy (e.g., pharmacy practice or the pharmacy laboratory). Nonetheless, it is believed that such biases have been equally distributed, and the purpose of the exercise was mostly to verify if there is diversity. Certainly, clinical pharmacy is just one possible area, and if focus would be put on another, results could eventually be quite different. This diversity may have implications in clinical practice and in public health in general. In fact, as faculties are spread out through the country, assuming that this diversity captured in clinical pharmacy exists in other areas, we may easily understand that some advanced community pharmacy services for example are easier to implement in some areas than others. Conversely, industry based on pharmaceutical development may preferably search candidates from elsewhere. Currently, there is widespread awareness of contextual differences between faculties, but if focus is placed on identifying tailored solutions, shared learning outcomes may be obtained.

Despite its merits, this initiative has also some limitations worth acknowledging. The fact that it is not a research project with a specific timeline and very clear-cut indicators makes implementation more difficult, and hard to monitor. Academics and practitioners often have different views, making debates challenging, which is not a limitation per se, but may hinder the progress of any changes intended. Perhaps the most important limitation, which is still potential, and is hoped to be possible to be overcome, is the inability to ensure that this initiative will be sustained in future mandates. It is anticipated that the way to solve this is by giving the EPP free wings to be able to fly independently in the future, maintaining the ability to align education and practice for the sake of a more qualified workforce, with or without political support.

The main take home messages from this experience are that all stakeholders should be involved in all phases of developing a competent workforce. Some countries are already reviewing their educational models for more efficient healthcare [21]. The traditional approach of academia developing their curricula without the engagement of practitioners from all areas of pharmaceutical sciences, current and future recipients of care and products, i.e., the public and other professionals also involved in the drug lifecycle, is no longer acceptable. International, national and regional societies should develop their own platforms and working methods for discussing the most suitable and adaptable formats to their culture, to achieve a competent workforce that is able to meet current societal demands.

## Figures and Tables

**Figure 1 pharmacy-08-00011-f001:**
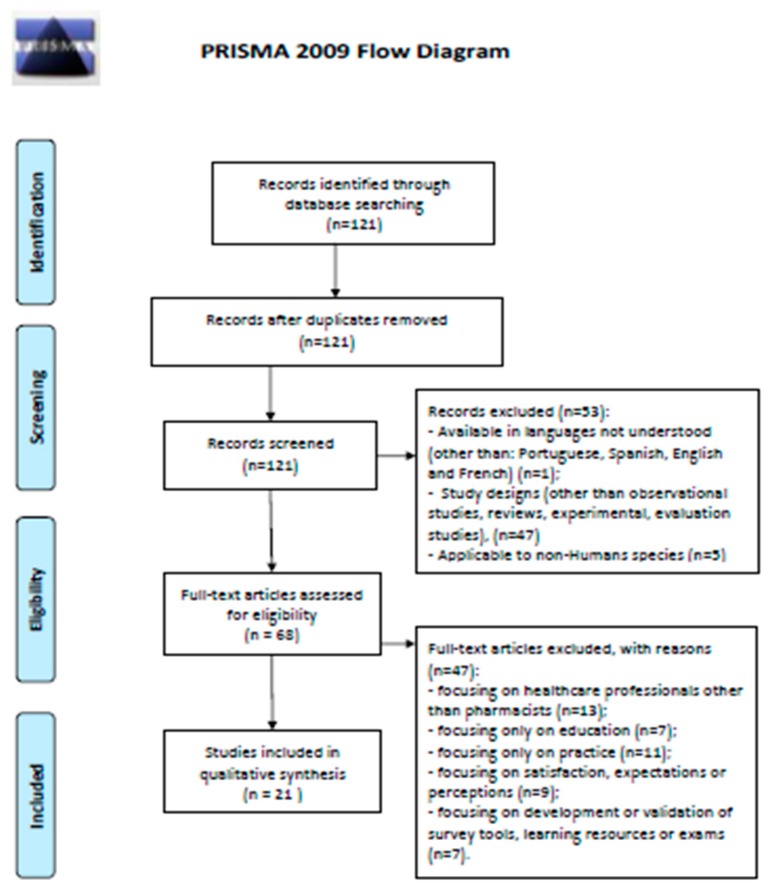
Preferred Reporting Items for Systematic Reviews and Meta-Analysis (PRISMA) 2009 Flow Diagram.

**Figure 2 pharmacy-08-00011-f002:**
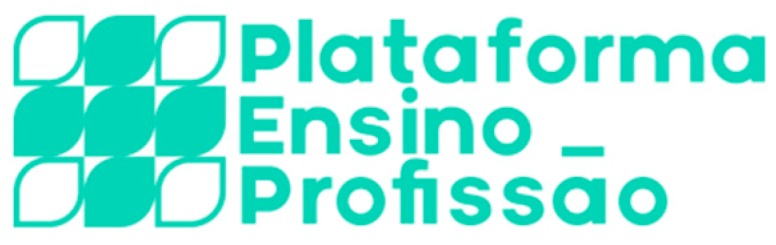
Logo of the Education and Practice Platform.

**Figure 3 pharmacy-08-00011-f003:**
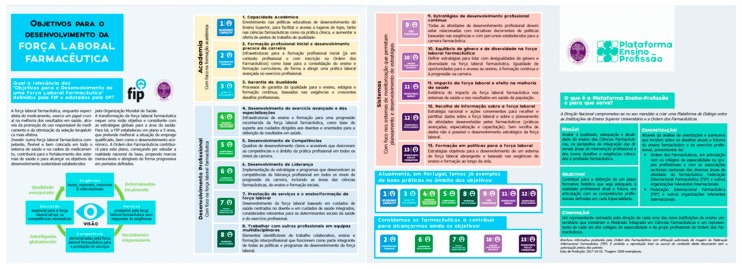
Brochure introducing the Education and Practice Platform (EPP) and the Workforce Development Goals (WDGs).

**Table 1 pharmacy-08-00011-t001:** Key Performance Indicators (KPIs) for the Education and Practice Platform (EPP).

Domain	KPI Set	Period of Monitoring
Workshop activities	Number of workshops held Proportion of invited participants present Satisfaction of participants (stratified per area of activity, i.e., academia vs practice)	2016–2021, with an internal evaluation in 2019
Involvement in international events	Number of events identified as particularly relevant for the EPP’s aims where work is presented	
Answering to global requests	Number of organizations/events/tasks with whom the EPP cooperated	
Answering to national needs	Creation of a common training framework for hospital internships (met/unmet) Creation of a common training framework for community pharmacy internships (met/unmet) Report by Boards of Pharmacy Practice describing their view on the advantages of these common training frameworks. This report will be made by area (hospital pharmacy and community pharmacy) and will include a section with the remaining areas, which will be structured as desired training delivered through undergraduate and postgraduate education. Development of a roadmap identifying postgraduate education available in all faculties, including those needed for less common areas of practice (e.g., research methods; pharmacogenetics) Establishment of a protocol between the nine higher education institutions and the PPS for Continuous Professional Development (CPD) (met/unmet)	

Legend: KPI = key performance indicator; EPP = Education and practice platform; PPS = Portuguese Pharmaceutical Society; CPD = Continuous Professional Development.

**Table 2 pharmacy-08-00011-t002:** Clinical pharmacy component of Pharmaceutical Sciences MSc degrees in nine institutions in Portugal.

Institution	# Semesters	# Contact Hours/Semester	# ECTS/Semester	% Practical Teaching
1	3	56	5	73
2	1	60	5	43
3	2	60	4	33
4	2	112	9	68
5	1	60	5	50
6	3	52	5	67
7	2	49	4	60
8	1	75	6	60
9	1	52	4	50

Legend: # = number; ECTS = European Credit Transfer System; % = Proportion. Note: optional topics were excluded from this analysis; only topics within the scope of clinical pharmacy, as defined by ESCP, were considered, which means that, e.g., pharmacology and pharmacotherapy are not included.

**Table 3 pharmacy-08-00011-t003:** Summary table about the common and diverse features in the MSc in Pharmaceutical Sciences offered by nine institutions in Portugal.

Core Areas	ECTS Conferred MSc	Area with Strongest Component	Proportion of Employed Graduates	Post-Graduate Offer	Particularities
Institution 1	300	Pharmaceutical Sciences	94.8%	6 MSc; 1 PG; 1 PhD	Various international partners Unique center for monitoring drug-herbal interactions
Institution 2	300	Pharmaceutical Sciences	99.4%	7 MSc; 3 PG; 5 PhD	Various partnerships with other faculties for PhD
Institution 3	300	Health Sciences	95.0%	0 MSc; 0 PG; 6 PhD	Various partnerships with other faculties for PhD
Institution 4	300	Biology and Biochemistry	94.9%	3 MSc; 0 PG; 0 PhD	Compulsory internships throughout the degree in various areas
Institution 5	300	Pharmaceutical Sciences	95.0%	4 MSc; 14 PG, 0 PhD	Summer internships throughout the degree in various areas
Institution 6	300	Pharmaceutical Sciences	98.1%	1 MSc; 0 PG; 1 PhD	Sole institution with growing number of students
Institution 7	300	Pharmaceutical Sciences	95.7%	0 MSc; 10 PG; 1 PhD	Teacher practitioner concept (integrated with local community). Interprofessional collaboration in teaching (e.g., clinical cases)
Institution 8	300	Pharmaceutical Sciences	Not Available	0 MSc; 0 PG; 0 PhD	Has own hospital to ensure internships
Institution 9	300	Pharmaceutical Sciences	95.0%	1 MSc; 0 PG; 1 PhD	International partnerships for PhD. Teaching organized in trimesters

Legend: ECTS = European Credit Transfer System; MSc = Master in Sciences; PG = post-graduation; PhD = Doctorate degree.

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
