# Peer review of "Development of a Platform to Align Education and Practice: Bridging Academia and the Profession in Portugal"

_pharmacy, 2020, doi:10.3390/pharmacy8010011_

Round 1

Reviewer 1 Report

The manuscript entitled "Development of a Platform to Align Education and Practice: bridging academia and the profession in Portugal" is very interesting, informative and well written. Therefore, in my opinion it can be accepted. However I have one comment:

Are similar projects to Education & Practice Platform being carried out in other countries? I suggest to mention about this in the Introduction.

Author Response

Thank you very much for your positive comments on our manuscript.

We have checked the spelling to correct for minor errors highlighted.

Regarding your question, to our knowledge there are no similar published initiatives developed in any country. Nonetheless, we do consider there might be unpublished work we are unfamiliar with and the main reason for this statement is that we believe such initiatives have mostly a policy character, hence making them very hard to publish. We have added such reflections on the discussion.

Reviewer 2 Report

Thank you for the opportunity to contribute to the peer review process for the original manuscript submission entitled 'Development of a Platform to Align Education and Practice: bridging academia and the profession in Portugal' (pharmacy-651531).

Global comments:

This reads more like a professional magazine report or a project outcome report rather than a scientific research article. It appears that the submission is a post priori consideration to disseminate information, rather than an a priori plan of research conduct. The methodology described is not as robust as one may expect in a scientific journal for a research endeavour, instead more akin to a quality improvement project or an organisational initiative. There is for example no systematic nature to the literature review, no formal consensus mechanism / delphi process or a priori methodology plan described as to the valid and reliable development of the EPP. Narrative reporting on entity mission vision & values statements, results of workshop activities, and logo development is not normally the expectant content of a scientific journal publication. So in this reviewers opinion there is much opportunity to revise the presentation of the material to better suite the medium in which it is being submitted. By their own admissions (line 308) this was not a research project, so it may be that this report may be more appropriate for an alternate knowledge transmission pathway.

The following are offered as specific suggestions to authors to further enhance the dissemination of their message to the journal audience, should the decision be to still target a scientific research journal format:
Page 1, Abstract, line 30 - suggest amending 'Unfitness' - whilst technically not incorrect it is very dichotomous (to 'fitness') when in fact fitness to practice as a concept is not so black and white - perhaps rephrase to 'Poor fitness' or 'Inadequate fitness' or 'Improper fitness' etc

line 44 - suggest amending to not capitalize 'key'

line 45 - suggest amending punctuation to '...events, and answering...'

line 46 - abbreviation FIP not defined prior to use

page 2, line 48 - suggest amending to utilize previously defined abbreviation '...of the PPS was created...'

Page 3, Introduction, line 61 - suggest amending from presumptive gender specific language here ('his activities'), as presumably male dominance is not the current status quo?

line 79 - suggest amending to take opportunity to define abbreviation here '...Society (PPS) initiated...'

line 84 - suggest amending to '...[12-15]. A revised version...'

line 85 - suggest rephrasing to '...has recently been...' (as 2018 is no longer 'just been')

line 87 - suggest amending to utilize previously defined abbreviation '...the PPS developed...'

Materials and Methods, line 94 - suggest amending to utilize previously defined abbreviation '...of an EPP was...'

lines 101-108 - there are many expected elements missing from this search strategy that limit the ability to repeat and interpret the findings of the review. For example, what were the date limiters of the search, was there a language component (English only, English & Portuguese, etc)(given results report exclusion of a Japanese then presumably there was no language restrictions to search?), why pubmed only as database searched, who reviewed & decided on articles to be included, etc? The authors are recommended to provide a more comprehensive documenting of the search strategy extending to:

Databases searched (and why), including database provider/platform (eg. OVID Medline, ProQuest PsycINFO, Ebsco CINAHL) Date search was conducted Search strategy: subject headings and keywords used, including whether terms were exploded, truncated, and how terms were combined
Years searched Filters used

The search process needs to be documented in enough detail throughout the process to ensure that it can be reported correctly in the manuscript, to the extent that all the searches of all the databases are reproducible.

Page 4, line 138 - suggest capitalize 'Table 1'

abbreviation BPP in Table 1 not defined (but then not used again in manuscript -therefore suggest just writing in full)
abbreviation CPD not defined prior to use in Table 1

page 5, line 142 - suggest incorporating a legend for table.

line 146, Results - suggest capitalize 'Table 2'. One would have reasonably expected more extensive reporting of the outcomes of the literature search. There should be reporting of:

Number of results retrieved for each search Total number of records  Duplicates identified Numbers pre-screening and post-screening

Accepting the authors may not have attempted/completed a systematic review, elements of reporting visually through a modified PRISMA type flow chart would be a reasonable inclusion.

Further the results here are reporting that 7 categories of exclusion were created - these should have been created a priori and included in methodology. Why were data from other professions excluded as not relevant for pharmacy? There is no specificity in description / criteria as to what was 'outdated' or 'low quality' and as such there are potential concerns of bias resultant.

line 152 - supplementary table 1 was not evident in submission (although supplementary table 2 is [on page 10, line 317]).

Page 6, line 199 - suggest amending to utilize previously defined abbreviation '...the ESCP, through...'

line 201 - suggest capitalize 'Table 2'

Page 7,line 204 - suggest add legend to table

line 210- suggest amending to utilize previously defined abbreviation '...of the WDGs, issued...'

Page 8, line 230 - suggest rephrasing to 'Further, as the...'

line 232 - suggest amending 'WDG' to 'WDGs' to be consistent with defined abbreviation and use elsewhere in manuscript

line 235 - was this sentiment applicable for pharmacists or more broadly for all professions? Said who? - suggest provide more explicit details.

line 236 - suggest amending to depersonalize 'Currently a recommendation is in place...'

line 239 - abbreviation 'ECTS' not defined prior to use

line 253 - suggest capitalize 'Table 3'

line 254 - suggest amending to 'MSc'

Page 9, Table 3 - suggest add legend

Discussion, line 257 - suggest amending to depersonalize '...which was believed had been...'

Line 258- suggest amending to depersonalize 'Additionally, it was intended...'

line 260 - suggest amending to utilize previously defined abbreviation '...bu the PPS in...'

line 281 - suggest amending to depersonalize 'Nonetheless, experience tells...'

Page 10, line 283 - suggest amending to depersonalize '...teams, all are...'

line 290 - suggest amending tense '...views were considered...practitioners, was crucial...'

line 292 - suggest amending to depersonalize '...example, the need is cited for...'

line 302 - suggest amending to depersonalize 'Nonetheless, it is believed...'

line 304 - suggest amending to depersonalize '...and if focus was on...'

line 305- suggest amending to depersonalize 'Currently there is widespread awareness of contextual differences between faculties, but focus will be...'

line 311 - suggest amending to depersonalize '...and it is hoped to...'

line 312 - suggest amending to depersonalize 'It is anticipated that the way to solve this is...'

Page 12, References, line 342 - review italicizing of journal title; also missing details of volume, pages, doi etc

Page 13, ref #8 - review font size for consistency

ref #9 - review capitalization of manuscript title, and abbreviation for journal title

ref #19 - review capitalization of manuscript title

ref #20 - review font sizing

ref #27 - review capitalization of manuscript title

Page 14, ref #30 - review font sizing

ref #36 -? review / update (as e-pub ahead of print from 2018)

Author Response

Point by point responses are provided in the attached document.

Reviewer 3 Report

Overall, a very interesting manuscript as it pertains to how to better bridge the gap of pharmacy education to practice in a developing country.  

Please see below my feedback

Major feedback

The Intro does not clearly convey the following: the overarching problem that your research is trying to solve, what do we know about the topic? What is the knowledge gap you
are filling?  Your aims.  So rework intro to address each of these things clearly.   Line 106-108 explains the main aim, it should not be in the materials and methods, but rather in the intro The methods needs to be reworked to include: Design, Setting,  Sample, etc.  For example line 146-150 in the results section has listed the exclusion and inclusion criteria of the literature searches Also the methods do not clearly define how the study will answer the primary aim " Based on the PICO acronym: Are there teaching experiences at the undergraduate level that better meet the needs of pharmacy practitioners" line 153 "Establishing the Platform" this should be in the discussion, the results should only capture the descriptive statistics, then present results in order of primary aim,
secondary aim, etc and it does not clearly follow that framework.  Line 191-193 should be part of the discussion  " future work...." not results The discussion typically should summarize the main most important result of the paper, discuss most important result in the context of the literature,  discuss implications of findings for clinical practice and/or public health, limitations of the study,  brief summary of main take home points line 247-248 " ....national education in pharmaceutical sciences" loses the audience because the paper was initially discussing how to better align pharmacy education with the profession.  Pharmaceutical sciences is completely different from pharmacy practice

Minor feedback

line 37 replace "as" with "a" line 44 not clear what the intent for the word "fora" is? could it be "forum"? if so make changes also to line 53,line 289 line 44 capitalization of "Key", no need for caps  line 46 remove "one on"  line 61 replace "his" with "him/her" or remove "his" as it reads now it makes it seem like only males are considered pharmacist line 66 replace "long-life" with "life-long" and same with line 92 line 70-72 ("moreover, the initially...) consider rewording? not sure what you are trying to convey  line 79 abbreviation of PPS, same for line 87 line 82 remove "for" line 84 remove "the current year" and replace with "Most recently" line 94-95 consider rewording not sure what you are trying to convey also line 94 "education and practice platform" already abbreviated line 107 "PICO" not previously defined

Author Response

Responses attached

Round 2

Reviewer 2 Report

Thank you for the opportunity to contribute again to the peer review process for the revised submission of the manuscript entitled "Development of a Platform to Align Education and Practice: bridging academia and the profession in Portugal (pharmacy-651531_R1)". The authors have extensively outlined their response to the review process, many new changes are evident in the revised manuscript in response to the editor’s and the three peer reviewer's recommendations, and the manuscript has evolved positively via this review process.

The following suggested amendments are recommended for consideration:

Page 2 of 15, Introduction, lines 65 & 76 - suggest substituting 'undergraduate' with 'entry level' - in academia the MSc is a post graduate qualification (as a Bachelor program would be the undergraduate degree) although it (MSc) can be the entry to practice qualification

line 71 - suggest amending to '...whereas those subsequently...'

line 89 - suggest amending to '...no single, universal solution...'

page 3 - lines 94-96 - review line spacing for consistency with rest of manuscript.

Materials & Methods, 2.1 - suggest depersonalize:
line 112 -'It was decided...'

line 113 - '...and a systematic review was not aimed for.'

line 114 - 'Nonetheless, the Preferred...(figure 1) was used.'

line 115 - 'A priori restrictions were not made on the... until it could be seen if...'

line 117 'Results were restricted to...'

Page 4, Figure 1 - suggest adding '(n=121)' to initial Identification box & '(n=21)' to the Included box. The numbers do not seem to make sense between the screened (n=121) box & the assessed for eligibility box (n=68) when the side box of exclusions is considered (as it totals 100). How did you get from 121 to 68? Then the full text exclusions box is missing final line of text

Page 6, Results, 3.1 - suggest review this paragraph as there is some duplicate reporting with details covered in Figure 1

Page 7, line 211 - reviewer query as to whether 'University of Monash' is correct? This reviewer was only able to identify a 'Monash University' in Australia - is this where the example came from?

Page 9, line 271 - suggest amend such that sentence does not start with an abbreviation: 'In 2004 CPD...'

References

Page 12, line 397 (#1) - suggest capitalize journal title

line 403 (#3) - does this journal title get abbreviated?

#4, #5, #6, #7 - suggest review capitalization of manuscript title for consistency with rest of list / style guide

line 419 (#9) - does this journal title get abbreviated?

Page 13, #24, #27 - suggest review capitalization of manuscript title

lines 462/3 (#26) - does this journal title get abbreviated?

#34 - capitalize 'British Columbia'

Page 14, line 501 - suggest leave space before supplementary table

Author Response

Comments and Suggestions for Authors

Thank you for the opportunity to contribute again to the peer review process for the revised submission of the manuscript entitled "Development of a Platform to Align Education and Practice: bridging academia and the profession in Portugal (pharmacy-651531_R1)". The authors have extensively outlined their response to the review process, many new changes are evident in the revised manuscript in response to the editor’s and the three peer reviewer's recommendations, and the manuscript has evolved positively via this review process.

We are pleased to read that reviewer 2 was satisfied by amendments made.

The following suggested amendments are recommended for consideration:

Page 2 of 15Introduction, lines 65 & 76 - suggest substituting 'undergraduate' with 'entry level' - in academia the MSc is a post graduate qualification (as a Bachelor program would be the undergraduate degree) although it (MSc) can be the entry to practice qualification

We thank the reviewer for the explanation and have changed the term accordingly. Also, given this correction, we opted to change in line 78 “undergraduate course” by “MSc course”.

line 71 - suggest amending to '...whereas those subsequently...'

Changed as suggested.

line 89 - suggest amending to '...no single, universal solution...'

Changed as suggested.

page 3 - lines 94-96 - review line spacing for consistency with rest of manuscript.

Edited as suggested.

Materials & Methods, 2.1 - suggest depersonalize:
line 112 -'It was decided...'

Changed as suggested.

line 113 - '...and a systematic review was not aimed for.'

Changed as suggested.

line 114 - 'Nonetheless, the Preferred...(figure 1) was used.'

Changed as suggested.

line 115 - 'A priori restrictions were not made on the... until it could be seen if...'

Changed as suggested.

line 117 'Results were restricted to...'

Changed as suggested.

Page 4Figure 1 - suggest adding '(n=121)' to initial Identification box & '(n=21)' to the Included box. The numbers do not seem to make sense between the screened (n=121) box & the assessed for eligibility box (n=68) when the side box of exclusions is considered (as it totals 100). How did you get from 121 to 68? Then the full text exclusions box is missing final line of text

We corrected the figure and pasted it ensuring no cuts to lines were made. Regarding the counts, we have corrected them for greater clarity (as some excluded for one reason were the same as for a second reason – first box – making it confusing)

Page 6Results, 3.1 - suggest review this paragraph as there is some duplicate reporting with details covered in Figure 1

We have simplified the paragraph.

Page 7, line 211 - reviewer query as to whether 'University of Monash' is correct? This reviewer was only able to identify a 'Monash University' in Australia - is this where the example came from?

Yes, the University is in Australia (although they also have a pole in Malaysia). One of the lecturers from Monash was invited to come to our workshop and discuss their teaching. We have added the location between brackets.

Page 9, line 271 - suggest amend such that sentence does not start with an abbreviation: 'In 2004 CPD...'

Changed as suggested.

References

Page 12, line 397 (#1) - suggest capitalize journal title

Changed

line 403 (#3) - does this journal title get abbreviated?

Abbreviated

#4, #5, #6, #7 - suggest review capitalization of manuscript title for consistency with rest of list / style guide

We have reviewed all journal titles so that they follow the recommendations for abbreviations, capitalisation and use of italics.

line 419 (#9) - does this journal title get abbreviated?

Abbreviated

Page 13, #24, #27 - suggest review capitalization of manuscript title

We have reviewed all journal titles so that they follow the recommendations for abbreviations, capitalisation and use of italics.

lines 462/3 (#26) - does this journal title get abbreviated?

Abbreviated

#34 - capitalize 'British Columbia'

Changed

Page 14, line 501 - suggest leave space before supplementary table

Page break added

Reviewer 3 Report

Thank you for addressing all the comments very clearly!

Author Response

We are very pleased to see that our answers satisfied reviewer 3 and thank him/her for the constructive criticism, making us achieve a better version.